# Latent Functional Maps:
# a spectral framework for representation alignment

**Marco Fumero**[*]
IST Austria
marco.fumero@ist.ac.at

**Marco Pegoraro**[*]
Sapienza, University of Rome
pegoraro@di.uniroma1.it

**Valentino Maiorca**[†]
Sapienza, University of Rome
maiorca@di.uniroma1.it

**Francesco Locatello**
IST Austria
francesco.locatello@ist.ac.at

**Emanuele Rodolà**
Sapienza, University of Rome
rodola@di.uniroma1.it

## Abstract

Neural models learn data representations that lie on low-dimensional manifolds, yet modeling the relation between these representational spaces is an ongoing challenge. By integrating spectral geometry principles into neural modeling, we show that this problem can be better addressed in the functional domain, mitigating complexity, while enhancing interpretability and performances on downstream tasks. To this end, we introduce a multi-purpose framework to the representation learning community, which allows to: (i) compare different spaces in an interpretable way and measure their intrinsic similarity; (ii) find correspondences between them, both in unsupervised and weakly supervised settings, and (iii) to effectively transfer representations between distinct spaces. We validate our framework on various applications, ranging from stitching to retrieval tasks, and on multiple modalities, demonstrating that Latent Functional Maps can serve as a swiss-army knife for representation alignment.

## 1 Introduction

Neural Networks (NNs) learn to represent high-dimensional data as elements of lower-dimensional latent spaces. While much attention is given to the model's output for tasks such as classification, generation, reconstruction, or denoising, understanding the internal representations and their geometry is equally important. This understanding facilitates reusing representations for downstream tasks [62, 14] and the comparison between different models [23, 39], thereby broadening the applicability of NNs, and understanding their structure and properties. Moreover, recent studies have shown that distinct neural models often develop similar representations when exposed to similar stimuli, a phenomenon observed in both biological [16, 25, 63] and artificial settings [38, 23, 39]. Notably, even when neural networks have different architectures, internal representations of distinct models can often be aligned through a linear transformation [61, 50]. This indicates a level of consistency in how NNs process information, highlighting the importance of exploring and understanding these internal representations.

---

[*]Equal Contribution
[†]Work done while visiting ISTA

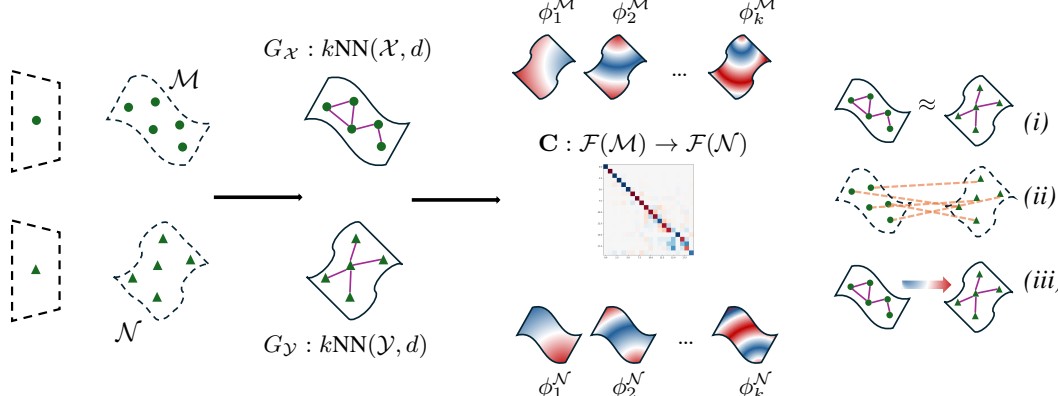

Figure 1: **Framework overview:** given two spaces $\mathcal{X}$, $\mathcal{Y}$ their samples lie on two manifolds $\mathcal{M}$, $\mathcal{N}$, which can be approximated with the KNN graphs $G_{\mathcal{X}}$, $G_{\mathcal{Y}}$. We can optimize for a *latent functional map $C$* between the eigenbases of operators defined on the graphs. This map serves as a map between functions defined on the two manifolds and can be leveraged for (i) comparing representational spaces, (ii) solving correspondence problems, and (iii) transferring information between the spaces.

In this paper, we shift our focus from characterizing relationships between samples in distinct latent spaces to modeling a map among function spaces defined on these representational spaces. To this end, we propose leveraging spectral geometry principles by applying the framework of *functional maps* [44] to the field of representation learning. Functional maps represent correspondences between function spaces on different manifolds, providing a new perspective on the problem of representational alignment. In this setting, many complex constraints can be easily expressed compactly [45]. For instance, as shown in Figure 1, the mapping ($C$) between two spaces $\mathcal{M}$ and $\mathcal{N}$ can be represented in the functional space as a linear map with a sparse structure.

Originally used in 3D geometry processing and more recently on graphs applications [60, 46], we extend this framework to handle arbitrary large dimensional latent spaces. Our proposed method, *Latent Functional Map* (LFM), is a flexible tool that allows (i) to compare distinct representational spaces, (ii) to find correspondences between them both in an unsupervised and weakly supervised way, and (iii) to transfer information between them. Our contributions can be listed as follows:

- We introduce the framework of Latent Functional Maps as a way to model the relation between distinct representational spaces of neural models.
- We show that LFM allows us to find correspondences between representational spaces, both in weakly supervised and unsupervised settings, and to transfer representations across distinct models.
- We showcase LFM capabilities as a meaningful and interpretable similarity measure between representational spaces.
- We validate our findings in retrieval and stitching tasks across different models, modalities and datasets, demonstrating that LFMs can lead to better performance and sample efficiency than other methods.

## 2 Related Work

**Similarity between latent spaces.** Comparing representations learned by neural models is of fundamental importance for a diversity of tasks [63], ranging from representation analysis to latent space alignment and neural dynamics. In order to do so, a similarity measure between different spaces must be defined [22]. This can range from functional similarity (matching the performance of two models) to similarity defined in representational space [23], which is where our framework falls in. A classical statistical method is Canonical Correlation Analysis (CCA) [18], known for its invariance to linear transformations. Various adaptations of CCA aim to enhance robustness, such as Singular

Vector Canonical Correlation Analysis (SVCCA) [48], or to decrease sensitivity to perturbations using methods like Projection-Weighted Canonical Correlation Analysis (PWCCA) [38]. Closely related to these approaches, Centered Kernel Alignment (CKA) [23] measures the similarity between latent spaces while ignoring orthogonal transformations. However, recent research [10] reveals that CKA is sensitive to local shifts in the latent space.

We propose to leverage LFMs as a tool to measure the similarity, or how much two spaces differ from an isometry w.r.t. to the metric that has been used to construct the graph.

**Representation alignment.** Complementary to measuring the similarity of distinct representational spaces, several methods have been proposed to optimize for a transformation to align them [1]. The concept of *latent space communication*, introduced by [39], builds on the hypothesis that latent spaces across neural networks, varying random seed initialization to architecture or even data modality, are intrinsically compatible. This notion is supported by numerous empirical studies [38, 30, 23, 6, 55, 3, 58, 27, 29, 4, 40, 9, 15, 8], with the phenomenon being particularly evident in large and wide models [51, 33]. The core idea is that relations between data points (i.e., distances according to some metric) are preserved across different spaces because the high-level semantics of the data are the same and neural networks learn to encode them similarly [19]. With this "relative representation", the authors show that it is possible to *stitch* [29] together model components coming from different models, with little to no additional training, as long as a partial correspondence of the spaces involved is known. Indeed, [26, 32, 35, 37] show that a simple linear transformation is usually enough to effectively map one latent space into another, measuring the mapping performance via desired downstream tasks.

With LFMs, we change the perspective from merely relating samples of distinct latent spaces to relating function spaces defined on the manifold that the samples approximate, showing that processing information in this dual space is convenient as it boosts performance while also being interpretable.

**Functional Maps.** The representation we propose is directly derived from the functional maps framework for smooth manifolds introduced in the seminal work by [44]. This pioneering study proposed a compact and easily manipulable mapping between 3D shapes. Subsequent research has aimed at enhancing this framework. For instance, [41] introduced regularization techniques to improve the informativeness of the maps, while [34] developed refinement methods to achieve more accurate mappings. The functional map framework has been extended as well outside the 3d domain, for example, in [60] and [17], who applied the functional framework to model correspondences between graphs, and in [46], who demonstrated its utility in graph learning tasks. In particular, they have shown that the functional map representation retains its advantageous properties even when the Laplace basis is computed on a graph.

Inspired by these advancements, our work leverages the functional representation of latent spaces of neural models. We demonstrate how this representation can be easily manipulated to highlight similarities and facilitate the transfer of information between different spaces, thereby extending the applicability of the functional maps framework to the domain of neural latent spaces.

## 3  Method

### 3.1  Background

This section provides the basic notions to understand the framework of functional maps applied to manifolds. We refer to [45] for a comprehensive overview.

Consider two manifolds $\mathcal{M}$ and $\mathcal{N}$ equipped with a basis $\mathbf{\Phi}^{\mathcal{M}}$ (respectively $\mathbf{\Phi}^{\mathcal{N}}$). Any squared integrable function $f : \mathcal{M} \to \mathbb{R}$ can be represented as a linear combination of basis functions $\mathbf{\Phi}_{\mathcal{M}}$: $f = \sum_i a_i \Phi_i^{\mathcal{M}} = \mathbf{a}\mathbf{\Phi}_{\mathcal{M}}$. Given a bijective correspondence $T : \mathcal{M} \to \mathcal{N}$ between points on these manifolds, for any real-valued function $f : \mathcal{M} \to \mathbb{R}$, one can construct a corresponding function $g : \mathcal{N} \to \mathbb{R}$ such that $g = f \circ T^{-1}$. In other words, the correspondence $T$ defines a mapping between two function spaces $T_F : \mathcal{F}(\mathcal{M}, \mathbb{R}) \to \mathcal{F}(\mathcal{N}, \mathbb{R})$. Such a mapping is *linear* [44] and can be represented as a matrix $\mathbf{C}$ such that for any function $f$ represented as a vector of coefficients $\mathbf{a}$, we have $T_F(\mathbf{a}) = \mathbf{C}\mathbf{a}$.

The functional representation is particularly well-suited for map inference (i.e., constrained optimization problems). When the underlying map $T$ (and by extension the matrix $\mathbf{C}$) is unknown, many natural constraints on the map become linear constraints in its functional representation. In practice, the simplest method for recovering an unknown functional map is to solve the following optimization problem:

$$\arg\min_{\mathbf{C}} ||\mathbf{CA} - \mathbf{B}||^2 + \rho(\mathbf{C}) \tag{1}$$

where $\mathbf{A}$ and $\mathbf{B}$ are sets of corresponding functions, denoted as *descriptors*, expressed in the bases on $\mathcal{M}$ and $\mathcal{N}$, respectively, and $\rho(\mathbf{C})$ represents additional constraints deriving from the properties of the matrix $\mathbf{C}$ [45]. When the manifolds $\mathcal{M}$ and $\mathcal{N}$ are approximately isometric and the descriptors are well-preserved by the (unknown) map, this procedure provides a good approximation of the underlying map. In cases where the correspondence $T$ is encoded as a permutation matrix $\mathbf{P}$, the functional map can be retrieved as $\mathbf{C} = \mathbf{\Phi}_{\mathcal{N}}^{\dagger}\mathbf{P}\mathbf{\Phi}_{\mathcal{M}}$ where $\mathbf{\Phi}_{\mathcal{M}}$ and $\mathbf{\Phi}_{\mathcal{N}}$ are the bases of the functional spaces $\mathcal{F}(\mathcal{M},\mathbb{R})$ and $\mathcal{F}(\mathcal{N},\mathbb{R})$, respectively, and $^{\dagger}$ denotes the Moore-Penrose pseudo-inverse.

## 3.2 Latent Functional Maps

### 3.2.1 Setting

We consider deep neural networks $f := f_1 \circ f_2 \circ ... f_n$ where each layer $f_i$ is associated to a representational space $\mathcal{X}_i$ corresponding to the image of $f_i$. In the following we're gonna drop the subscript $i$, when the layer considered is clear form the context. We assume that elements $x \in \mathcal{X}$ are sampled from a latent manifold $\mathcal{M}$. Considering pairs of spaces $(\mathcal{X}, \mathcal{Y})$, and corresponding manifolds $\mathcal{M}, \mathcal{N}$ our objective is to characterize the relation between them by mapping the space of functions $\mathcal{F}(\mathcal{M})$ to $\mathcal{F}(\mathcal{N})$. Our framework can be summarized in three steps, which we are going to describe in the following sections: (i) how to represent $\mathcal{M}$ from a sample estimate $X$, by building a graph in the latent space $\mathcal{X}$ (ii) how to encode any known preserved quantity between the two spaces by defining a set of descriptor function for each domain (iii) optimizing for the latent functional map between $\mathcal{M}$ and $\mathcal{N}$. An illustrative description of our framework is depicted in Figure 1.

**Building the graph.** To leverage the geometry of the underlying manifold, we model the latent space of a neural network building a symmetric k-nearest neighbor (k-NN) graph [31]. Given a set of samples $X = \{x_1, \ldots, x_n\}$, we construct an undirected weighted graph $G = (X, E, \mathbf{W})$ with nodes $X$, edges $E$, and weight matrix $\mathbf{W}$. The weight matrix is totally characterized by the choice of a distance function $d : \mathcal{X} \times \mathcal{X} \mapsto \mathbb{R}$.Suitable choices for $d$ include the $L2$ metric or the angular distance. Edges $E$ are defined as $E = \{(x_i, x_j) \in X \times X \mid x_i \sim_{\mathrm{k}} x_j \text{ or } x_j \sim_{\mathrm{k}} x_i\}$, where $x_i \sim_{\mathrm{k}} x_j$ indicates that $x_j$ is among the $k$ nearest neighbors of $x_i$. The weight matrix $\mathbf{W} \in \mathbb{R}_{\geq 0}^{n \times n}$ assigns a weight $\omega(x_i, x_j) = \alpha(d(x_i, x_j))$ to each edge $(x_i, x_j) \in E$, and $\mathbf{W}_{i,j} = 0$ otherwise, with $\alpha$ being some monotonic function. Next, we define the associated weighted graph Laplacian operator $\mathcal{L}_G = \mathbf{I} - \mathbf{D}^{-1/2}\mathbf{W}\mathbf{D}^{-1/2}$, where $\mathbf{D}$ is the diagonal degree matrix with entries $\mathbf{D}_{i,i} = \sum_{j=1}^{n} \mathbf{W}_{i,j}$. $\mathcal{L}_G$ is a positive semi-definite, self-adjoint operator [57]), therefore, it admits an eigendecomposition $\mathcal{L}_G = \mathbf{\Phi}_G \mathbf{\Lambda}_G \mathbf{\Phi}_G^T$, where $\mathbf{\Lambda}_G$ is a diagonal matrix containing the eigenvalues, and $\mathbf{\Phi}_G$ is a matrix whose columns are the corresponding eigenvectors. The eigenvectors form an orthonormal basis for the space of functions defined on the graph nodes (i.e., $\mathbf{\Phi}_G^T \mathbf{\Phi}_G = \mathbf{I}$). We give comprehensive details on the choice of $k$ in Appendix A.1.

**Choice of descriptors.** To define the alignment problem between pair of spaces $\mathcal{X}, \mathcal{Y}$ we will introduce the notion of *descriptor function*. We define as descriptor function any real valued function $f : G \mapsto \mathbb{R}^k$ defined on the nodes of the graph. Informally descriptors should encode the information which is *shared* between $\mathcal{X}$ and $\mathcal{Y}$, either explicitly or implicitly . We categorize descriptor functions into supervised, weakly supervised and unsupervised descriptors. For the former we assumed to have access to a partial correspondence between the domains $\mathcal{X}, \mathcal{Y}$, defined as a bijective mapping $\Gamma_S : \mathcal{A}_{\mathcal{X}} \mapsto \mathcal{A}_{\mathcal{Y}}$ where $\mathcal{A}_{\mathcal{X}} \subset \mathcal{X}, \mathcal{A}_{\mathcal{Y}} \subset \mathcal{Y}$. Then descriptors takes the form of distance functions $d_{\mathcal{X}} : \mathcal{X} \times \mathcal{A}_{\mathcal{X}} \mapsto \mathbb{R}^+, d_{\mathcal{Y}} : \mathcal{Y} \times \Gamma_S(\mathcal{A}_{\mathcal{X}}) \mapsto \mathbb{R}^+$. As an example considering the geodesic distance (shortest path distance on the k-NN graph) from each node of the graph to the nodes in the anchor set, is an instance of supervised descriptor. In the case of weakly supervised descriptor, we assume to have access to an equivalence relation defined $\Gamma_W : \mathcal{A}_{\mathcal{X}} \times \mathcal{A}_{\mathcal{Y}} \mapsto \{0, 1\}$. Example of these are multi-to-multi mappings, like mappings between labels. Unsupervised descriptors are quantities that

depends on the topology and the geometry of the graph itself and they don't rely on any pre-given correspondence or equivalence relation. An example of this is the Heat Kernel Signature [53] node descriptor, based on heat diffusion over the manifold. We give examples of descriptors and ablation in Appendix C.2.

**Computing LFMs**. We now describe the optimization problem to compute a latent functional map between $\mathcal{X}$ and $\mathcal{Y}$. We model each space using a subset of training samples $X = \{x_1, \ldots, x_n\}$ and $Y = \{y_1, \ldots, y_n\}$ and build the k-NN graphs $G_X$ and $G_Y$ from these samples, respectively. For each graph, we compute the graph Laplacian $\mathcal{L}_G$ and derive the first $k$ eigenvalues $\Lambda_G$ and eigenvectors $\Phi_G = [\phi_1, \ldots, \phi_k]$, which serve as the basis for the function space defined on the latent spaces.

Given the set of descriptors $\mathbf{F}_{G_X} = [f_1^{G_X}, \ldots, f_{n_f}^{G_X}]$ and $\mathbf{F}_{G_Y} = [f_1^{G_Y}, \ldots, f_{n_f}^{G_Y}]$, we consider the optimization problem defined in Equation 1 and incorporate two regularizers which enforce commutativity for the Laplacian and the descriptors, respectively, with the map $\mathbf{C}$:

$$\arg \min_{\mathbf{C}} \|\mathbf{C}\hat{\mathbf{F}}_{G_X} - \hat{\mathbf{F}}_{G_Y}\|_F^2 + \alpha \rho_{\mathcal{L}}(\mathbf{C}) + \beta \rho_f(\mathbf{C}) \tag{2}$$

where $\hat{\mathbf{F}}_G = \Phi_G^T \mathbf{F}_G$ are the spectral coefficients of the functions $\mathbf{F}_G$, $\rho_{\mathcal{L}}$ and $\rho_f$ are the Laplacian and descriptor operator commutativity regularizers respectively, akin to [41]. We give full details on the regularizers in Appendix A. Once we have solved the optimization problem defined in Equation 2, we refine the resulting functional map $\mathbf{C}$ using the spectral upsampling algorithm proposed by [34], as detailed in Appendix A.3.

## 3.3 LFMs as a similarity measure

Once computed, the functional map $\mathbf{C}$ can serve as a measure of similarity between spaces. The reason is that for isometric transformations between manifolds, the functional map is volume preserving (see Thm 5.1 in [44]), and this is manifested in orthogonal $\mathbf{C}$. By defining the inner product between functions $h_1, h_2 \in \mathcal{F}(M)$ as $\langle h_1, h_2 \rangle = \int_{\mathcal{M}} h_1(x) h_2(x) \mu(x)$, it holds that $\langle h_1, h_2 \rangle = \langle \hat{h}_1, \hat{h}_2 \rangle$ when the map preserves the local area, where $\hat{h}$ denotes the functional representation of $h$. In other words, when the transformation between the two manifolds is an isometry, the matrix $\mathbf{C}^T \mathbf{C}$ will be diagonal. By measuring the ratio between the norm of the off-diagonal elements of $\mathbf{C}^T \mathbf{C}$ and the norm of the full matrix, we can define a measure of similarity $sim(X, Y) = 1 - \frac{\|\text{off}((\mathbf{C}^T\mathbf{C})\|_F^2}{\|\mathbf{C}^T\mathbf{C}\|_F^2}$. In Appendix A.4 we prove that this similiarity measure is a proper distance on the space of functional maps, allowing to compare measurements from collections of spaces. Furthermore, this quantity is interpretable; the first eigenvector of $\mathbf{C}^T \mathbf{C}$ can act as a signal to localize the area of the target manifold where the map has higher distortion [43], as we show in the experiment in Figure 3a.

## 3.4 Transfering information with LFM

The map $\mathbf{C}$ computed between two latent spaces can be utilized in various ways to transfer information from one space to the other. In this paper, we focus on two methods: (i) Expressing arbitrary points in the latent space as distance function on the graph and transferring them through the functional domain; (ii) Obtaining a point-to-point correspondence between the representational spaces from the LFM (see the first step in Appendix A.3) , starting from none to few known pairs, and leverage off-the-shelf methods to learn a transformation between the spaces. Additional strategies could be explored in future work.

**Space of Functional Coefficients.** The space of functional (or spectral) coefficients offers an alternative representation for points in the latent space $\mathcal{X}$. Using the equation $\hat{f}_G = \Phi_G^T f_G$, any function $f_G \in \mathcal{F}(G, \mathbb{R})$ can be uniquely represented by its functional coefficients $\hat{f}_G$. We leverage this property to represent any point $x \in \mathcal{X}$ as a distance function $f_d \in \mathcal{F}(G, \mathbb{R})$ from the set of points $X_G$, which correspond to the nodes of the graph $G$. The functional map $\mathbf{C}$ between two latent spaces $\mathcal{X}$ and $\mathcal{Y}$ aligns their functional representations, enabling the transfer of any function from the first space to the second. This functional alignment can be used similarly to the method proposed by [39] to establish a shared space where the representational spaces $\mathcal{X}$ and $\mathcal{Y}$ are aligned.

**Finding correspondences.** As explained in Section 3, the functional map $\mathbf{C}$ represents the bijection $T$ in a functional form. [44] demonstrated that this bijection can be retrieved as a point-to-point

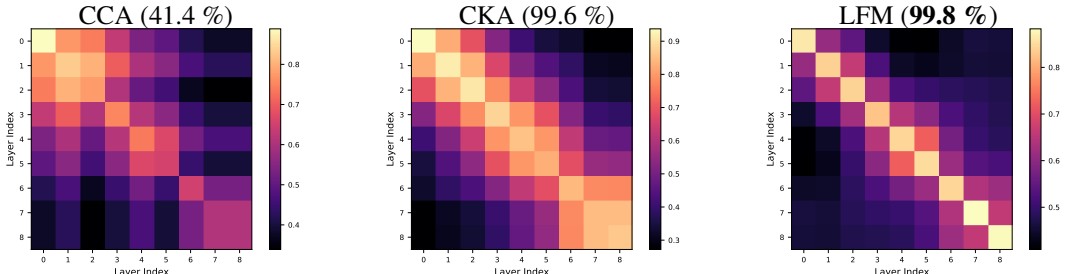

Figure 2: **Similarity across layers** Similarity matrices between internal layer representations of `CIFAR10` comparing our LFM-based similarity with the CCA and CKA baselines, averaged across 10 models. For each method, we report the accuracy scores for matching the corresponding layer by maximal similarity.

map by finding the nearest neighbor for each row of $\mathbf{\Phi}_{G_Y}\mathbf{C}$ in $\mathbf{\Phi}_{G_X}$. This process can be efficiently implemented and scaled up using kd-tree structures or approximation strategies [21, 20]. Starting from an empty set or few known correspondences (anchors) between the two spaces $\mathcal{X}$ and $\mathcal{Y}$, we can extend the correspondence to the entire set of nodes $X$ and $Y$. This extended set of anchors can then be used to fit an arbitrary transformation between the latent spaces, for example an orthogonal mapping [32]. In our experiments, we demonstrate that by using a very small number of anchors (typically $\leq 50$), we can retrieve optimal transformations that facilitate near-perfect stitching and retrieval tasks.

## 4 Experiments

In this section, we present a series of experiments designed to evaluate the effectiveness of the Latent Functional Map (LFM) framework. We explore its application across various tasks, including similarity evaluation, stitching, retrieval performance, and robustness to perturbations in latent space. By comparing LFM to existing methods under different conditions, we aim to demonstrate its versatility and superior performance in aligning and analyzing neural network representations. Additional ablations and qualitative visualizations on the choice of distance metric to construct the graph and descriptors selection are reported in the Appendix C.

### 4.1 Analysis

We demonstrate the benefits of using latent functional maps for comparing distinct representational spaces, using the similarity metric $sim(X, Y)$ defined in Section 3.3

**Experimental setting** In order to validate experimentally if LFMs can serve as a good measure of similarity between distinct representational spaces, we run the same sanity check as [23]. We train 10 CNN models (the architecture is depicted in Appendix B.1) on the `CIFAR-10` dataset [24], changing the initialization seed. We compare their inner representations at each layer, excluding the logits and plot them as a similarity matrix, comparing with Central Kernel Alignment (CKA) measure [23] and Canonical Correlation Analysis (CCA) [18, 48]. We then measure the accuracy of identifying corresponding layers across models and report the results comparing with CKA and CCA as baselines. For CCA, we apply average pooling on the spatial dimensions to the embeddings of the internal layers, making it more stable numerically and boosting the results for this baseline compared to what was observed in [23].

**Result analysis** Figure 2 shows that our LFM-based similarity measure behaves correctly as CKA does. Furthermore, the similarities are less spread around the diagonal, favoring a slightly higher accuracy score in identifying the corresponding layers across models.

While CKA (Centered Kernel Alignment) is a widely used similarity metric in deep learning, recent research by [10] has shown that it can produce unexpected or counter-intuitive results in certain situations. Specifically, CKA is sensitive to transformations that preserve the linear separability of

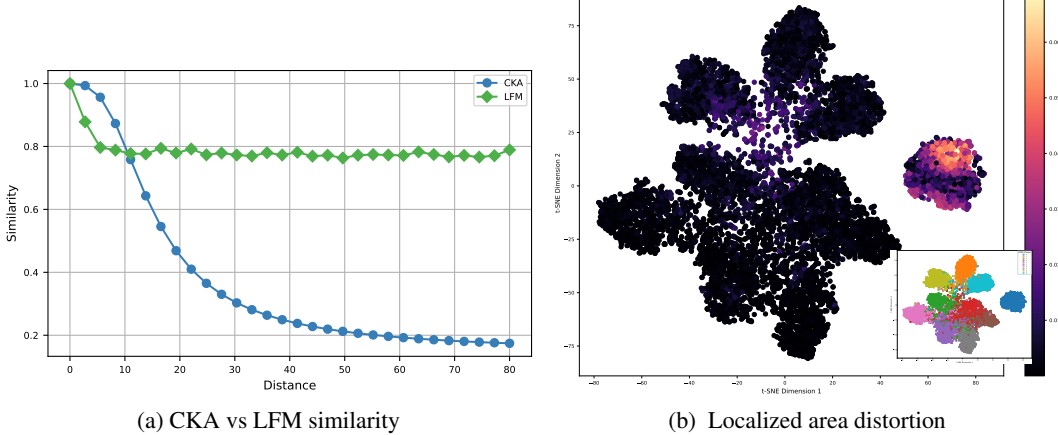

(a) CKA vs LFM similarity                    (b) Localized area distortion

Figure 3: **Robustness of LFM similarity** *Left:* Similarity scores as a function of perturbation strength: while the CKA baseline degrades, our LFM similarity scores are robust to perturbations that preserve linear separability of the space. *Right:* Visualization of area distortion of the map by projecting the first singular component of the LFM in the perturbed space: the distortion localizes on the samples of the perturbed class, making LFM similarity interpretable.

two spaces, such as local translations. Our proposed similarity measure is robust to these changes and demonstrates greater stability compared to CKA.

**Experimental setting**    We compute the latent representations from the pooling layer just before the classification head for the `CIFAR10` train and test sets. Following the setup in [10], we train a Support Vector Machine (SVM) classifier on the latent representations of the training samples to find the optimal separating hyperplane between samples of one class and others. We then perturb the samples by translating them in a direction orthogonal to the hyperplane, ensuring the space remains linearly separable. We measure the CKA and LFM similarities as functions of the perturbation vector norm, as shown in Figure 3a. In the accompanying plot on the right, we visualize the area distortion of the map by projecting the first singular component of the LFM **C** into the perturbed space and plotting it on a 2d TSNE [56] projection of the space.

**Result Analysis**    We start by observing that when the latent space is perturbed in a way that still preserves its linear separability, it should be considered identical from a classification perspective, as this does not semantically affect the classification task. Figure 3a shows that while CKA degrades as a function of perturbation intensity, the LFM similarity remains stable to high scores. To understand this difference, we can visualize the area distortion as a function of the samples by projecting the first singular component of **C** onto the perturbed space. In Figure 3b, we use t-SNE [56] to project the perturbed samples and the distortion function into 2D. The visualization reveals that distortion is localized to the samples corresponding to the perturbed class.

### 4.2    Zero-shot stitching

In latent communication tasks, a common challenge is combining an encoder that embeds data with a decoder specialized in a downstream task, such as classification or reconstruction—this process is often referred to as *stitching*. Previous works, like [29] and [2], have used trainable linear projections, known as stitching layers, to enable the swapping of different network components. However, [39] introduced the concept of zero-shot stitching, where neural components can be marged without any additional training procedure. It's important to note that while [39] still required to trained a decoder module once for processing relative representations before stitching could be performed, our method is fully zero-shot, with no need for additional training. In the following experiments, stitching is conducted without any training or fine-tuning of the encoder or decoder, adhering strictly to a *zero-shot fashion*.

**Experimental Setting**    We consider four pre-trained image encoders (see Appendix B.2 for details) and stitch their latent spaces to perform classification using a Support Vector Machine (SVM) on five

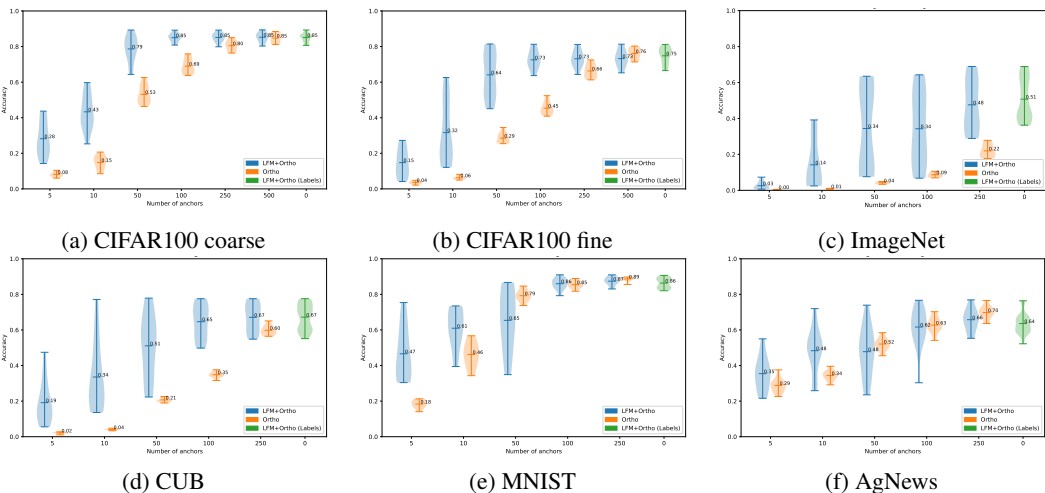

Figure 4: **Stitching results.** Accuracy performance in stitching between image encoders trained on 6 datasets, comparing orthogonal transformation (Ortho) and LFM+Ortho at varying anchor counts. Also shown is LFM+Ortho (Labels), which uses the dataset labels instead of anchors. Results are presented with mean accuracy values reported on each box.

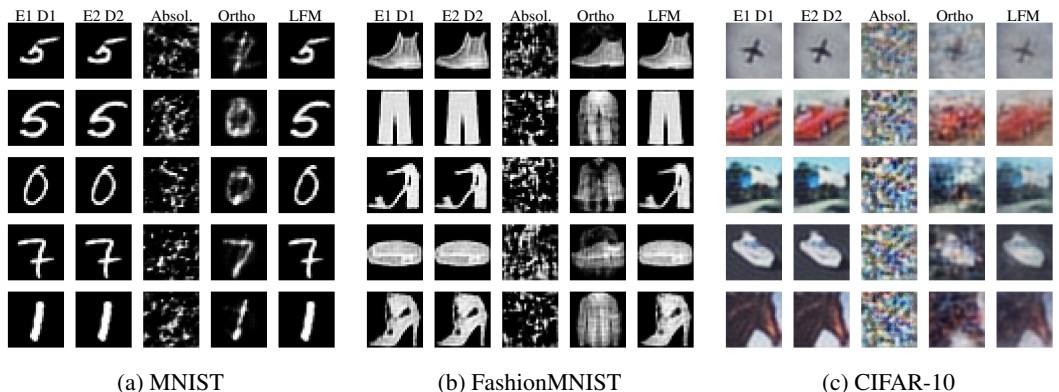

Figure 5: **Qualitative results on stitching.** We combine the encoder decoder modules of two convolutional autoencoder (*E1 D1*, *E2 D2*) using three different approaches: no transformation (*Absol.*), the orthogonal transformation from [32] (*Ortho*), the orthogonal transformation augmented with LFM (*LFM*).

different datasets: CIFAR100 [24] with coarse and fine-grained labelling, ImageNet [11], Caltech-UCSD Birds-200-2011 (CUB) [59], MNIST [12], AgNews [65]. These datasets were chosen to encompass both large-scale, complex datasets like ImageNet and a variety of modalities, including text and images. To evaluate the effectiveness of integrating the functional map, we extend the correspondences to determine an orthogonal transformation [32] between the latent spaces. For each encoder, we compute a graph of 3,000 points with 300 neighbors per node. We optimize the problem in Equation 2 using the first 50 eigenvectors of the graph Laplacian and consider two different descriptors: the distance functions defined from the anchors (LFM+Ortho) and the labels (LFM+Ortho (Labels)). For each dataset class, the latter provides an indicator function with 1 if the point belongs to the class and 0 otherwise. This descriptor does not require any anchor as input, representing a pioneering example of stitching requiring no additional information beyond the dataset.

**Result Analysis** Figure 4 shows the accuracy results for all possible combinations of encoder stitching. The latent functional map (LFM+Ortho) consistently outperforms other methods across all datasets, particularly with a low number of anchors. Even without anchors, using label descriptors (LFM+Ortho (Labels)) achieves strong performance across different labeling schemes. The orthogonal transformation computed directly from the anchors (Ortho) only matches LFM's performance

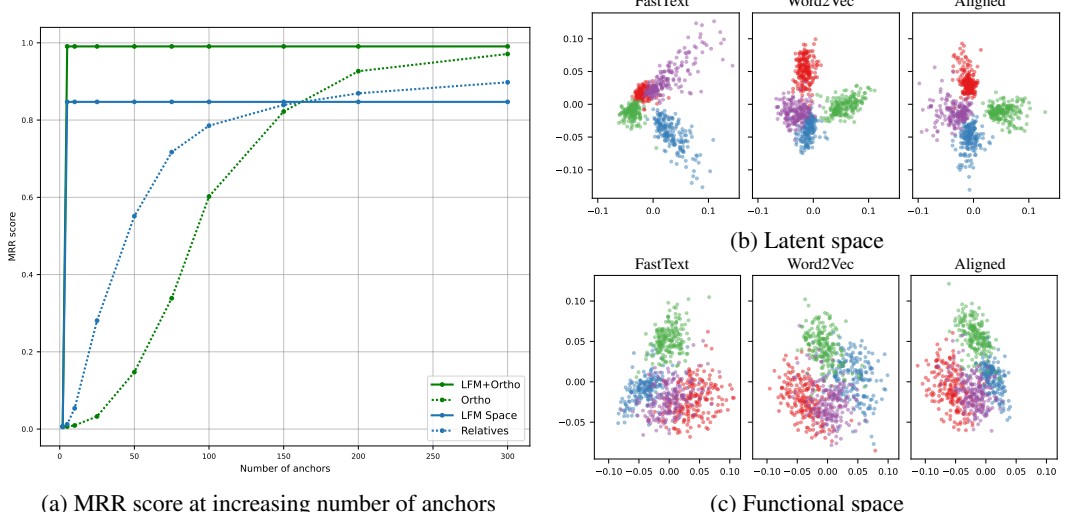

(a) MRR score at increasing number of anchors

(b) Latent space

(c) Functional space

Figure 6: **Retrieval of word embeddings.** We compare the retrieval performance of the functional map framework with state-of-the-art models as the number of anchors increases. The left panel shows the Mean Reciprocal Rank (MRR) across different numbers of anchors. The right panels depict the first two components of PCA for a subsample of the latent space (b) and the functional space (c), both before and after alignment using the functional map.

when more than 50 anchors are used, at which point LFM's effectiveness is constrained by the number of eigenvectors. Comparing the CIFAR100 coarse and fine-grained labeling, the fine-grained task is more complex due to the larger number of classes, making the estimation of the map more challenging. This complexity is especially evident when aligning self-supervised vision models with classification-based ones (e.g., DINO vs. ViT), where the training strategies differs.

This experiment shows that the latent functional map is highly effective when few anchors are available ($\leq 50$). It significantly enhances performance in zero-shot stitching tasks, outperforming direct orthogonal transformations at low or no anchor counts. This suggests that the latent functional map method provides a robust means of aligning latent spaces with minimal correspondence data, making it a valuable tool for tasks requiring the integration of independently trained models.

### 4.2.1 Qualitative example

In Figure 5, we present qualitative experiments on the MNIST, FashionMNIST [64], and CIFAR-10 datasets, focusing on visualizing the reconstructions of stitched autoencoders. For these experiments, we trained convolutional autoencoders using two different seeds and stitched their encoder and decoder modules together. As a baseline, we used no transformation (Absol.), where the latent representation from the first encoder is directly input into the second decoder. We then compared this baseline with the orthogonal transformation from [32] (Ortho) and its LFM-augmented version (LFM). For both our method and [32], we used 10, 10 and 50 correspondences for the MNIST, FashionMNIST, and CIFAR-10 datasets, respectively. Our method consistently produced superior reconstruction quality compared to both the baseline and the method from [32].

### 4.3 Retrieval

We extend our analysis to the retrieval task, where we look for the most similar embedding in the aligned latent space.

**Experimental Setting** We consider two different English word embeddings, FastText [5] and Word2Vec [36]. Following the approach of [39], we extract embeddings of 20K words from their shared vocabulary using pre-trained models. We use 2K random corresponding samples to construct the k-NN graphs and evaluate the retrieval performance on the remaining 18K word embeddings. We test two settings in our experiments:

1. Aligning functional coefficients (LFM Space).

2. Computing an orthogonal transformation using the correspondences obtained by the functional map (LFM+Ortho).

For this experiment, we construct k-NN graphs with a neighborhood size of 300 and compute the functional map using the first 50 eigenvectors. We evaluate the methods' performance using the Mean Reciprocal Rank (MRR), as detailed in Appendix B.4. Our functional map methods are compared with the method proposed by [39] (Relatives) and the orthogonal transformation method proposed by [32] (Ortho). We choose to fit the same orthogonal transformation on top of the correspondence found by LFM to make the comparison the fairest possible, although in principle any off-the-shelf methods could be used to estimate the transformation once the new correspondence is found. We illustrate this versatility in Table 3 of the Appendix, where LFM is combined with other methods in the same task.

**Result Analysis**  Figure 6 shows the performance of these methods as the number of anchors increases. Numerical results are detailed in Table 2 in the Appendix. The functional map significantly improves performance with just 5 anchors, achieving an MRR of over 0.8. As the number of anchors increases, the performance of competing methods improves but still falls short of FMAP+Transform at 300 anchors, which reaches an MRR of 0.99. Interestingly, the performance of the functional map methods does not improve beyond 5 anchors, suggesting that this number of anchors is sufficient to achieve an optimal functional map between the spaces. These findings are further supported by Table 3 in the Appendix, where LFM consistently achieves superior MRR scores, particularly with fewer anchors, outperforming other baselines such as [29]. In Appendix C.3, we analyze how the results improve as the number of eigenvectors used to compute the functional map increases. Notably, the MRR score drastically increases to over 0.6 with more than 25 eigenvectors.

These results confirm that the latent functional map is a valuable tool in settings with little knowledge about correspondences. It significantly enhances retrieval performance with a minimal number of anchors, making it an efficient approach for aligning latent spaces. Moreover, its performance can be improved using a higher number of eigenvectors. In particular, our framework comprises at the same time (i) an interpretable similarity measure, (ii) a way to find correspondences, and (iii) solving for an explicit mapping between different spaces in a single framework, differently from [39] and [32] which attempt to solve just the latter implicitly.

## 5   Conclusions

In this paper, we explored the application of latent functional maps (LFM) to model and analyze the relationships between different latent spaces. Our approach leverages the principles of spectral geometry, providing a robust framework for comparing and aligning representations across various models and settings. By extending the functional map framework to high-dimensional latent spaces, we offer a versatile method for comparing distinct representational spaces, finding correspondences between them in both unsupervised and weakly supervised settings (few or no anchors) and transferring information across different models.

**Limitations and future work**  The performance of LFM can be sensitive to the number of eigenvectors used, as observed in some of our experiments (see Appendix C.3). Finding the optimal number of eigenvectors without extensive experimentation remains an open problem. The current framework assumes a correspondence between the domains, and is not directly adapted in order to deal with partial mappings, where just a subset of one domain can be mapped to other. Extending the results in [49, 47] to our neural representation setting, is a promising way to overcome the full correspondence assumption. Future research can further explore the potential of LFM in other domains and tasks, and modalities. The versatility of the functional representation can be further explored to define new ad-hoc constraints and regularizers for different settings. In particular, ts performance in fully unsupervised settings, while promising is still an area requiring further research and improvement. In conclusion, the Latent Functional Map framework offers a novel and effective approach to understanding and utilizing neural network representations, promising significant advancements in both theoretical and practical aspects of machine learning.

## Acknowledgments and Disclosure of Funding

MF is supported by the MSCA IST-Bridge fellowship which has received funding from the European Union's Horizon 2020 research and innovation program under the Marie Skłodowska-Curie grant agreement No 101034413. ER and VM are supported by the PNRR MUR project PE0000013-FAIR. MP is supported by the Sapienza grant "Predicting and Explaining Clinical Trial Outcomes", prot. RG12218166FA3F13.

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

# A    Latent Functional Maps

## A.1    Building the graph

In the following section we give more details about how to construct the graph in the latent space in order to approximate the manifold domain from a sample estimate.

Throughout the paper, we assume the eigenvalues (and corresponding eigenvectors) are sorted in non-descending order $0 = \Lambda_1 \leq \Lambda_2 \leq \cdots \leq \Lambda_n$. One may consider a subset of eigenvectors, namely those associated with the $k$ smallest eigenvalues, to compactly approximate a graph signal, employing techniques akin to Fourier analysis. As demonstrated in multiple recent works [54, 7], the eigenvalues and eigenvectors of the graph Laplacian associated with a k-NN graph approximate the weighted Laplace-Beltrami operator, placing us in a setting similar to the original one of [44]. In particular in [7] it has been derived optimal bounds for the choices of $k$, bounds based on the dimensionality of the underlying manifold and the number of sampled points. In our experiments we choose our parameter $k$ according to these estimates. For an estimate of manifold dimensionality we use the latent space dimensionality, which correspond to an upper bound to it. In practice, we typically set $k = 300$ with a graph in the order of few thousands nodes.

## A.2    Additional Regularizers

In Equation 2, we improve the computation of the functional map by incorporating two additional regularizers: Laplacian commutativity and descriptor operator commutativity. Both regularizers exploit the preservation of linear functional operators $\mathbf{S}^G : \mathcal{F}(G, \mathbb{R}) \to \mathcal{F}(G, \mathbb{R})$, enforcing that the functional map $\mathbf{C}$ commutes with these operators: $\|\mathbf{S}_i^G \mathbf{C} - \mathbf{C}\mathbf{S}_i^{G_X}\|_F = 0$.

The Laplacian commutativity regularizer, first introduced by [44], is formulated as:

$$\rho_{\mathcal{L}}(\mathbf{C}) = \|\mathbf{\Lambda}_{G_Y}\mathbf{C} - \mathbf{C}\mathbf{\Lambda}_{G_X}\|_F^2 \tag{3}$$

where $\mathbf{\Lambda}_G$ represents the diagonal matrices of eigenvalues. This regularizer ensures that the functional map $\mathbf{C}$ preserves the spectral properties of the Laplacian.

The descriptor operator commutativity regularizer, introduced by [41], extracts more detailed information from a given descriptor, resulting in a more accurate functional map even with fewer descriptors. The formulation of this regularizer is as follows:

$$\rho_f(\mathbf{C}) = \sum_i \|\mathbf{S}_i^{G_Y}\mathbf{C} - \mathbf{C}\mathbf{S}_i^{G_X}\|_F^2 \tag{4}$$

where $\mathbf{S}_i^G = \mathbf{\Phi}_G^T(f_i^G)\mathbf{\Phi}_G$ are the descriptor operators.

## A.3    Spectral refinement

Once we have solved the optimization problem defined in Equation 2, we refine the resulting functional map $\mathbf{C}$ using the spectral upsampling algorithm proposed by [34]. We start by considering the matrix $\mathbf{C}$ as the submatrix of size $k \times k$ of the current LFM estimate. Then we apply iteratively the following two steps:

1. Compute the pointwise map $T$ encoded as matrix $\mathbf{\Pi}$ from the current LFM estimate $\mathbf{C}$ via solving: $\arg\min_y \|\mathbf{C}(\mathbf{\Phi}_{G_Y}^{:,y}) - \mathbf{\Phi}_{G_X}^{:,x}\|_2 \ \forall x \in X$, where $\mathbf{\Phi}^{:,x}$ correspond to the $x$-th column of the matrix

2. Set $\mathbf{C}_{k+1} = \mathbf{\Phi}_{G_X}^{:,k}{}^T \mathbf{\Pi}\mathbf{\Phi}_{G_Y}^{:,k}$, where $\mathbf{\Phi}^{:,k}$ correspond to the $k$-th column of the matrix $\mathbf{\Phi}$.

## A.4    LFM similarity properties

In this section we will demonstrate that the LFM similarity property $sim(X, Y) = 1 - \frac{\|\text{off}((\mathbf{C}^T\mathbf{C})\|_F^2}{\|(\mathbf{C}^T\mathbf{C})\|_F^2}$, defined in Section 3.3 is a metric, as it satisfies, positiveness, simmetry and triangle inequality.

Namely, for a pairs of spaces $X, Y$:

$$sim(X, Y) \geq 0 \tag{5}$$

$$sim(X, Y) = sim(Y, X) \tag{6}$$

$$sim(X, Z) \leq sim(X, Y) + sim(Y, Z) \tag{7}$$

Since $\|\text{diag}(\mathbf{C}^T\mathbf{C})\|_F^2 + \|\text{off}(\mathbf{C}^T\mathbf{C})\|_F^2 = \|(\mathbf{C}^T\mathbf{C})\|_F^2$ we will consider $sim(X, Y) = sim(X, Y) = \frac{\|\text{diag}(\mathbf{C}^T\mathbf{C})\|_F^2}{\|(\mathbf{C}^T\mathbf{C})\|_F^2}$ for simplicity. The former property is trivial as the ration between two norms is positive. For proving simmetry we have that:

$$sim(X, Y) = \frac{\|\text{diag}(\mathbf{C}^T\mathbf{C})\|_F^2}{\|(\mathbf{C}^T\mathbf{C})\|_F^2} \tag{8}$$

$$sim(X, Y) = \frac{\|\text{diag}(\mathbf{\Phi}_X^T\mathbf{\Phi}_Y)\|_F^2}{\|(\mathbf{\Phi}_X^T\mathbf{\Phi}_Y)\|_F^2} \tag{9}$$

for the numerator $\|(\mathbf{\Phi}_X^T\mathbf{\Phi}_Y)\|_F$ we trivially have $diag(\mathbf{\Phi}_X^T\mathbf{\Phi}_Y) = diag(\mathbf{\Phi}_Y^T\mathbf{\Phi}_X)$. For the denominator:

$$\text{Since :} \|\mathbf{\Phi}\|_F = \sqrt{\sum_{i,j} |\mathbf{\Phi}_{i,j}|^2} = \sqrt{Tr(\mathbf{\Phi}^T\mathbf{\Phi})} \tag{10}$$

We have that:

$$\|\mathbf{\Phi}_X^T\mathbf{\Phi}_Y\|_F^2 \stackrel{?}{=} \|\mathbf{\Phi}_Y^T\mathbf{\Phi}_X\|_F^2 \tag{11}$$

$$Tr(\mathbf{\Phi}_X^T\mathbf{\Phi}_Y)^T(\mathbf{\Phi}_X^T\mathbf{\Phi}_Y) \stackrel{?}{=} Tr(\mathbf{\Phi}_Y^T\mathbf{\Phi}_X)^T(\mathbf{\Phi}_Y^T\mathbf{\Phi}_X) \tag{12}$$

$$Tr(\mathbf{\Phi}_Y^T\mathbf{\Phi}_X\mathbf{\Phi}_X^T\mathbf{\Phi}_Y) = Tr(\mathbf{\Phi}_X^T\mathbf{\Phi}_Y\mathbf{\Phi}_Y^T\mathbf{\Phi}_X) \tag{13}$$

where the last equality is verified for the Trace operator being invariant under cyclic permutations.

For proving triangle inequality we will assume that the matrices of eigenvectors are full rank (i.e. all eigenvectors are considered). We have to prove the following:

$$1 - \frac{\|\text{diag}(\mathbf{\Phi}_X^T\mathbf{\Phi}_Z)\|_F^2}{\|(\mathbf{\Phi}_X^T\mathbf{\Phi}_Z)\|_F^2} \leq 2 - \left(\frac{\|\text{diag}(\mathbf{\Phi}_X^T\mathbf{\Phi}_Y)\|_F^2}{\|(\mathbf{\Phi}_X^T\mathbf{\Phi}_Y)\|_F^2} + \frac{\|\text{diag}((\mathbf{\Phi}_Y^T\mathbf{\Phi}_Z)\|_F^2}{\|(\mathbf{\Phi}_Y^T\mathbf{\Phi}_Z)\|_F^2}\right) \tag{14}$$

$$\frac{\|\text{diag}(\mathbf{\Phi}_X^T\mathbf{\Phi}_Z)\|_F^2}{\|(\mathbf{\Phi}_X^T\mathbf{\Phi}_Z)\|_F^2} \geq \left(\frac{\|\text{diag}(\mathbf{\Phi}_X^T\mathbf{\Phi}_Y)\|_F^2}{\|(\mathbf{\Phi}_X^T\mathbf{\Phi}_Y)\|_F^2} + \frac{\|\text{diag}((\mathbf{\Phi}_Y^T\mathbf{\Phi}_Z)\|_F^2}{\|(\mathbf{\Phi}_Y^T\mathbf{\Phi}_Z)\|_F^2}\right) - 1 \tag{15}$$

Using the fact that the Frobenius norm is invariant under multiplication by orthogonal matrices the denominator of each term in Eq14 becomes: $\|(\mathbf{\Phi}_X^T\mathbf{\Phi}_Y)\|_F^2 = \|\mathbf{\Phi}_X\|_F^2 = Tr[\mathbf{\Phi}_X^T\mathbf{\Phi}_X] = Tr[Id] = N$. Then:

$$\frac{\|\text{diag}(\mathbf{\Phi}_X^T\mathbf{\Phi}_Z)\|_F^2}{N} \geq \left(\frac{\|\text{diag}(\mathbf{\Phi}_X^T\mathbf{\Phi}_Y)\|_F^2}{N} + \frac{\|\text{diag}((\mathbf{\Phi}_Y^T\mathbf{\Phi}_Z)\|_F^2}{N}\right) - 1 \tag{16}$$

Since $\|\text{diag}(\mathbf{X}_X^T\mathbf{Y})\|_F^2 \leq \|(\mathbf{X}_X^T\mathbf{Y})\|_F^2$ for any $\mathbf{X}, \mathbf{Y}$ then the RHS of Eq. 16 will always be smaller than the LHS.

# B  Experimental details

## B.1  Architecture Details

All non-ResNet architectures are based on All-CNN-C [52]

## B.2  Pre-trained models

In Section 4.2 we used four pretrained models: 3 variations of [13] ('google-vit-base-patch16-224', 'google-vit-large-patch16-224', 'WinKawaks-vit-small-patch16-224') and the model proposed by [42] ( 'facebook-dinov2-base').

|               Tiny-10               |
|-------------------------------------|
| $3 \times 3$ conv. 16-BN-ReLu $\times 2$ |
| $3 \times 3$ conv. 32 stride 2-BN-ReLu |
| $3 \times 3$ conv. 32-BN-ReLu $\times 2$ |
| $3 \times 3$ conv. 64 stride 2-BN-ReLu |
| $3 \times 3$ conv. 64 valid padding-BN-ReLu |
| $1 \times 1$ conv. 64-BN-ReLu |
| Global average pooling |
| Logits |

Table 1

## B.3 Parameters and resources

In all our experiments we used gpu rtx 3080ti and 3090. In order to compute the eigenvector and functional map on a graph of 3k nodes we employ not more than 2 minutes.

## B.4 Mean Reciprocal Rank (MRR)

Mean Reciprocal Rank (MRR) is a commonly used metric to evaluate the performance of retrieval systems [39]. It measures the effectiveness of a system by calculating the rank of the first relevant item in the search results for each query.

To compute MRR, we consider the following steps:

1. For each query, rank the list of retrieved items based on their relevance to the query.

2. Determine the rank position of the first relevant item in the list. If the first relevant item for query $i$ is found at rank position $r_i$, then the reciprocal rank for that query is $\frac{1}{r_i}$.

3. Calculate the mean of the reciprocal ranks over all queries. If there are $Q$ queries, the MRR is given by:

$$\text{MRR} = \frac{1}{Q} \sum_{i=1}^{Q} \frac{1}{r_i} \tag{17}$$

Here, $r_i$ is the rank position of the first relevant item for the $i$-th query. If a query has no relevant items in the retrieved list, its reciprocal rank is considered to be zero.

MRR provides a single metric that reflects the average performance of the retrieval system, with higher MRR values indicating better performance.

## C Additional Results

In this section, we provide additional results and ablation studies that further support and expand upon the experiments presented in Section 4.

## C.1 Functional maps structure

In Section 3.2.1, we have shown that latent graphs can be constructed using different distance metrics. While angular distance was primarily used in our experiments, we now highlight its effectiveness and the resulting functional maps.

In Figure 7, we present a visualization of the structure of the functional map in a synthetic setting. The experiment was conducted as follows: given an input set $X$ consisting of test embeddings extracted from MNIST, we aimed to observe the degradation of the functional map structure as the space is perturbed. The perturbation involved an orthogonal transformation combined with additive Gaussian noise at increasing levels. In the first row, the functional maps were computed from k-nearest neighbor (knn) graphs using the angular distance metric, while in the second row, knn graphs were constructed using the L2 distance metric. Below each functional map, the LFM similarity score and MRR retrieval scores are displayed.

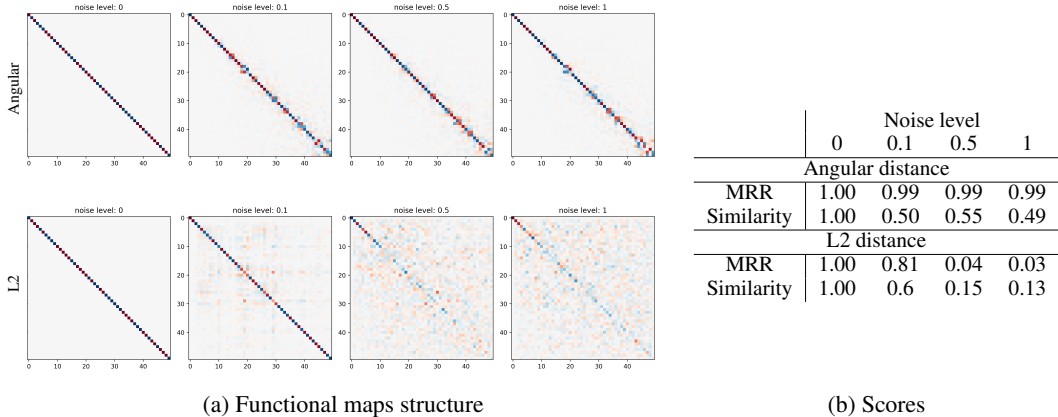

| | Noise level | | | |
|---|---|---|---|---|
| | 0 | 0.1 | 0.5 | 1 |
| Angular distance | | | | |
| MRR | 1.00 | 0.99 | 0.99 | 0.99 |
| Similarity | 1.00 | 0.50 | 0.55 | 0.49 |
| L2 distance | | | | |
| MRR | 1.00 | 0.81 | 0.04 | 0.03 |
| Similarity | 1.00 | 0.6 | 0.15 | 0.13 |

(a) Functional maps structure          (b) Scores

Figure 7: **Functional maps structure at increasing level noise.** Given a set of MNIST embeddings, we plot the degradation of the functional map structure as the space is perturbed. We compare the graph built with two different metrics (Angular, L2) and report MRR and LFM similarity score.

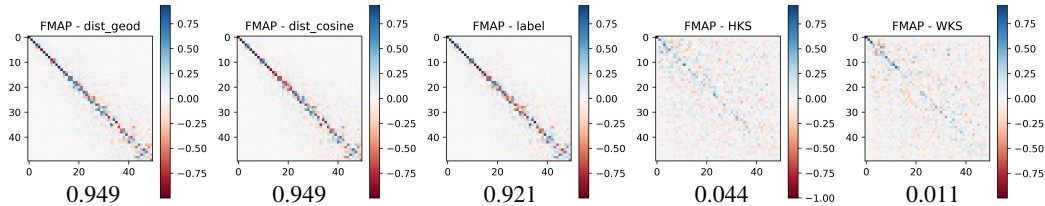

Figure 8: **Descriptors ablation.** We compare the latent functional maps computed on MNIST using different descriptors. For each map we report the MRR score.

We observed that (i) when noise is absent (first column), the two spaces are isometric, and the functional map is diagonal, (ii) constructing the graph with the cosine distance metric is more robust to increasing noise, and (iii) the LMF similarity score correlates with the MRR retrieval metric, indicating that more structured functional maps reflect better alignment between spaces.

## C.2 Ablation on the choice of descriptors

In Figure 8, we performed an ablation study on the choice of descriptors, comparing supervised descriptors (geodesic distance and cosine distance using 10 correspondences), weakly supervised descriptors (label descriptor), and fully unsupervised descriptors (heat kernel signature [53] and wave kernel signature [53]).

To conduct this study, we performed a retrieval task on the test embeddings of two convolutional autoencoders trained on MNIST, which differed by their parameter initialization. We visualized the structure of the functional map and reported the performance in terms of mean reciprocal rank (MRR), observing the following: (i) Geodesic and cosine descriptors performed best (ii) The geodesic distance (shortest path) is a good choice as it is agnostic of the metric chosen to build the initial graph, yet provides the same result as using the metric itself; (iii) The structure of the functional map reflects the performance of retrieval.

## C.3 Retrieval

In Table 2, we report the numerical results for the experiment in Figure 6 adding more transformations from the method of [32]: orthogonal (Ortho), linear (Linear) and affine (Affine). From the value in the table, we can see that all the methods that involve the latent functional map (LFM) saturate at 5 anchors, reaching top performance. In Figure 9, we show how the performance of the latent functional map methods depends on the number of eigenvectors used to compute the map. In particular, we

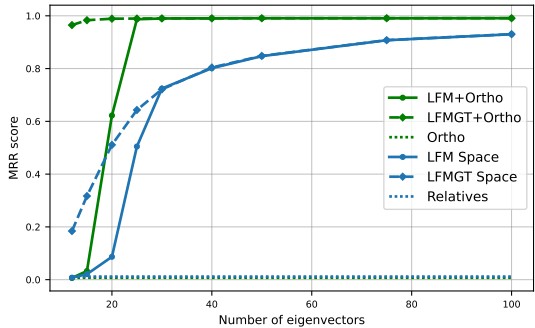

Figure 9: **Retrieval of word embeddings for increasing number of eigenvectors.** We report the MMR score for the methods in Figure 6 with fixed number of anchors (5) and increasing number of eigenvectors.

Table 2: **MRR Score for the retrieval of word embeddings.** We report the value of the results depicted in Figure 6 adding more kind transformation between spaces (Orthogonal, Linear and Affine).

| Method | Number of anchors | | | | | | | | | |
|---|---|---|---|---|---|---|---|---|---|---|
| | 2 | 5 | 10 | 25 | 50 | 75 | 100 | 150 | 200 | 300 |
| Relatives | 0.01 | 0.01 | 0.05 | 0.28 | 0.55 | 0.72 | 0.79 | 0.84 | 0.87 | 0.90 |
| Ortho | 0.01 | 0.01 | 0.01 | 0.03 | 0.15 | 0.34 | 0.60 | 0.82 | 0.93 | 0.97 |
| Linear | 0.01 | 0.01 | 0.01 | 0.05 | 0.26 | 0.49 | 0.66 | 0.77 | 0.74 | 0.01 |
| Affine | 0.01 | 0.01 | 0.01 | 0.04 | 0.19 | 0.45 | 0.64 | 0.81 | 0.89 | 0.95 |
| LFM Space | 0.01 | 0.85 | 0.85 | 0.85 | 0.85 | 0.85 | 0.85 | 0.85 | 0.85 | 0.85 |
| LFM+Ortho | 0.01 | 0.99 | 0.99 | 0.99 | 0.99 | 0.99 | 0.99 | 0.99 | 0.99 | 0.99 |
| LFM+Linear | 0.01 | 0.99 | 0.99 | 0.99 | 0.99 | 0.99 | 0.99 | 0.99 | 0.99 | 0.99 |
| LFM+Affine | 0.01 | 0.99 | 0.99 | 0.99 | 0.99 | 0.99 | 0.99 | 0.99 | 0.99 | 0.99 |

Table 3: **MRR Score for the retrieval of CUB.** We report the results on the CUB including the additional baseline Procrustes [29].

| Method | Number of anchors | | |
|---|---|---|---|
| | 5 | 50 | 250 |
| Relatives | 0.01 | 0.07 | 0.14 |
| Procrustes | 0.00 | 0.06 | 0.32 |
| Ortho | 0.00 | 0.06 | 0.33 |
| Linear | 0.00 | 0.04 | 0.18 |
| LFM+Procrustes | **0.07** | **0.24** | 0.33 |
| LFM+Ortho | 0.05 | **0.24** | **0.34** |
| LFM+Linear | 0.04 | 0.16 | 0.22 |

notice that the performance drastically increases at 25 eigenvectors, reaching the same score when using the functional map computed from the ground truth correspondences (LFMGT).

### C.3.1 Extended comparison

In Table 3 we added a comparison with multiple baselines: Relative [39], Ortho [32], Linear[28], Procrustes [29]. With the same foundation models and settings used in Section 4.3, we perform a retrieval task on the Caltech-UCSD Birds-200-2011 (CUB) dataset. We demonstrate that LFM is consistently superior in performance and can be used on top of any method which computes an explicit mapping between spaces.

### C.4 Additional qualitative results

In Figure 10 we show additional qualitative results on the MNIST, FMNIST, Cifar10 datasets, akin to the experiment in Figure 5 in the main manuscript.

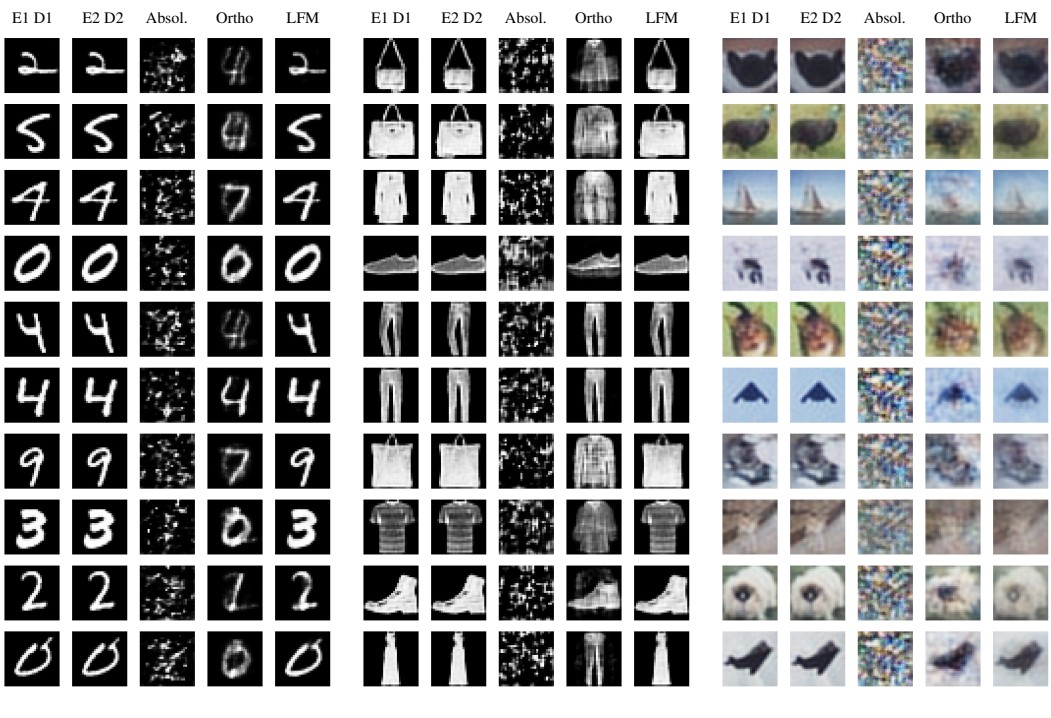

(a) MNIST       (b) FashionMNIST       (c) CIFAR-10

Figure 10: **Additional qualitative results on stitching.**

