# OpenReview forum: "Latent Functional Maps: a spectral framework for representation alignment"
_NeurIPS.cc/2024/Conference — NeurIPS 2024 poster_

### Official Review · Reviewer_yAi6 · 2024-06-27

**Soundness:** 3
**Presentation:** 2
**Contribution:** 3
**Rating:** 7
**Confidence:** 5

**Summary:**

The paper demonstrates the possibility to use the functional map tool on the embedding space of neural networks. The idea is that the embedding space of a neural network is usually lay on a low dimensional manifold, and networks that are trained for the same tasks even with different architecture result in similar manifolds in the embedding spaces. It was suggested in the paper to use the functional map as a tool for representation transfer for various tasks. The authors showed that using the functional maps improves over previous methods on several benchmarks (similarity measures, stitching, and word retrieval).

**Strengths:**

The paper presents a novel and elegant idea to utilize the functional map framework for neural representation transfer. In general, the paper is written clearly. In addition, the existing experiment shows superiority in comparison to previous methods.

**Weaknesses:**

There are several concerns regarding this paper that should be addressed by the authors before upgrading the rank. First, it is unclear why there is a need for least squares for computing the functional maps as the correspondence is known by construction. Why not calculate it directly? It is more accurate and does not require the zoomout post-processing. If the authors believe otherwise, they should add an ablation study showing it and explain why this is the case. Additionally, even if it was decided to calculate it via least squares, why the chosen descriptors were used the way they are - as there is no guarantee for their consistency in the two latent spaces. The authors should explain this important point, and may do an ablation study regarding the appropriate descriptors.
Second, In general, I would like to see more experiments (or at least similar experiments on more datasets) as in other papers in the field [1]. Additionally, in the stitching experiment, to build the graph the authors used 3000 points with 300 neighbors per point, in that case calculating geodesic distances is almost similar to calculating Euclidean distances in the latent space, which might impair the descriptors. Moreover, the method was not compared to the other baselines. The retrieval experimental setting is unclear.
Small issues: I think there is a mix up when using the notation X, Y, M and N (e.g. see caption of the overview figures).
[1]  Maiorca, Valentino, et al. "Latent Space Translation via Semantic Alignment." Advances in Neural Information Processing Systems 36 (2024).

**Questions:**

see above sections.

**Limitations:**

The authors should add the known limitations of the functional maps, such as the setting of full and partial manifolds [2, 3, 4]. Additionally, experiments that would significantly strengthen the claims are those that would show that the method presents the same limitations as those used in functional maps for 3D analysis. E.g an experiment showing the part-to-full manifold limits (e.g. training a network with part of the data that another network was trained on, and showing a slanted-diagonal structure of the functional map).
[2] Litany, O., et al.. "Fully spectral partial shape matching." In Computer Graphics Forum, vol. 36, no. 2, pp. 247-258. 2017.
[3] Rodolà, E., et al.. "Partial functional correspondence." In Computer graphics forum, vol. 36, no. 1, pp. 222-236. 2017.
[4] Bracha, A., et al.. "On Partial Shape Correspondence and Functional Maps." arXiv preprint arXiv:2310.14692 (2023).

---

> ### Author Rebuttal · Authors · 2024-08-07
>
> We thank the reviewer for their feedback and suggestions. We are gonna address their questions and concerns in the following, and remain available for any additional questions.
>
> **Know the correspondence**
>
> We fully agree with the reviewer that when the correspondence is known there is no need to optimize for the functional map C. We would like to clarify however, that the correspondence between the two domains is not (fully) available in most of our experiments. In general we would like to address both the cases in which the correspondence is known (for example measuring similarity between projections of the same samples in different networks, when inputs are available) and settings in which are not as stitching and retrieval tasks.
>
> In the experiment proposed (retrieval and stitching) our goal is to recover a good correspondence starting from very  few or coarse (labels) correspondences between the domains..
>
> We were glad to see however that the optimization recovered very good estimates of the ground truth functional maps.
>
>
> **Geodesic distance descriptor**
>
> We remark that the geodesic distance descriptor on the graph is computed just w.r.t to the available correspondences (anchors) on each node of the graph. Therefore the descriptor just encodes the anchors correspondence as distance functions, which is instead provided explicitly to fit the transformation in [22].
> We chose to employ the geodesic distance descriptor as it is agnostic to the metric used to build the graph (in this case cosine distance). Therefore the geodesics (shortest paths) on the graph will be a discretization of the distance metric used to build the knn graph, and are expressed just w.r.t. to the anchors.We clarified this in the paper.
> In general we explored different strategies for the choice of descriptor functions on top of the KNN graph, and choose the geodesic distance on the graph as it is agnostic of the choice of the metric used to construct the KNN graph  and provides a good choice in terms of performance. We included an ablation experiment on the choice of descriptor in the PDF of the general answer.
>
>
> **Experiments and baselines**
>
> We included many additional experiments in the general answers on diverse and large scale datasets. In Figure 1 we test stitching on diverse datasets (ImageNet,  MNIST, AGNews, DBPedia14, CUB) to validate the performance of LFM on multiple settings, including  large scale, complex, datasets (Imagenet).
>
> **Partiality limitation**
>
> We agree that the partiality case must be still explored and will include it in the limitation section.
>
> **Notation**
>
> We fixed the notation in the introduction. We thank the reviewer for spotting this.

---

> > ### Comment · Reviewer_yAi6 · 2024-08-11
> > **Authors made a nice effort.**
> >
> > The authors' made an effort addressing our concerns regarding the computation of the FM, the geodesic descriptor, and the limited experiments. The explanations are clear and satisfactory. We hope that the motivation for the FM computation and the process of construction geodesic descriptors would be included in paper.

---

> > > ### Author Response · Authors · 2024-08-11
> > > **Response to Reviewer comment**
> > >
> > > We thank the reviewer for their response and suggestions.
> > >
> > > We will include the discussion on the computations of the FM, the geodesic descriptor, and the experiments performed in the rebuttal period in the main paper.
> > >
> > > Since the reviewer has expressed satisfaction with our rebuttal, we kindly ask for consideration to adjust their score accordingly, as also mentioned in their initial review.
> > >
> > > We remain available for any further inquiries during the remainder of the discussion period.

---

### Official Review · Reviewer_9utW · 2024-07-09

**Soundness:** 3
**Presentation:** 2
**Contribution:** 3
**Rating:** 5
**Confidence:** 4

**Summary:**

This paper proposes using the functional maps paradigm for comparing and aligning the latent spaces of different neural architectures, possibly trained with different setups: initialization, different datasets, noise, etc.

The main contribution of this paper is to view the latent space as a Riemannian manifold, and comparing between such spaces is modeled as a correspondence problem. The authors leverage the functional maps paradigm prominent in geometry processing that effectively uses the spectral geometry of manifolds to obtain a compact and accurate representation of the map.

The authors demonstrate their idea on 3 (mostly synthetic) setups: (1.) Using F-Maps as a measure to compare between spaces (Sec 4.3), (2.) Stitching latent spaces (Sec 5.2) and (3.) Retrieval using word embeddings. The experiments demonstrate that the functional paradigm is quite good when the number of anchors (ground truth correspondences) is less making a case for the effective use of spectral geometry in such applications

**Strengths:**

- The core idea is interesting i.e. to view a latent space as a manifold and then use spectral geometry and F-Maps to compare multiple such spaces
- I found the experiments making a good proof of concept

**Weaknesses:**

- The submission builds considerably on the ideas introduced in [29] and it is important to highlight and compare the key conceptual differences (aside from just "using" F-Maps)
- The success of this approach depends on the availability of good anchors and label-based desriptors. Designing these descriptors for 3D shapes (HKS/WKS/SHOT) was straightforward. It is unclear from the submission how these descriptors can be generalized in the setting of representation spaces?
- Figure 2 is misleading if the colorbars are not the same.
- I found the baselines used for comparison to be rather limited. Why not compare with [18, 22, 25, 27, 20, 13]? To this aid, it is important to show some reconstruction examples similar to Figure 4 in [29] and compare with it
- As a general comment, the experimental section seems very hurried. Figures 3 and 5 need a more descriptive caption explaining what is to be learned there?
- It would help to make a figure/experiment showing the functional map matrices. While it is acceptable to digest that "similar objects have similar representations" and that "global distances are preserved", I am not fully convinced that this translates into an exact near-isometry model for latent representations. A good evidence would be to show F-Map matrices in different settings. To this aid, there could be some adjustments in the laplacian and descriptor commutativity constraints in Eq 2 and I find it a bit disappointing to not anything new here.

**Questions:**

- What is the significance of comparing LFmaps + Ortho with only Ortho?
- Why is the variance in the Fine Labelling case (4b) more than (4a)?

**Limitations:**

I do not see any direct negative social impact. Overall: I think this is a submission with some interesting findings and the use of functional maps in this setting is quite refreshing. However, I lean negatively on (1.) The lack of good baselines (2.) Stronger evidence on the applicability of functional maps (3.) Lack of clear communication on: progress over [29] and the experimental section not being fully self-contained.

---

> ### Author Rebuttal · Authors · 2024-08-07
>
> We thank the reviewer for their feedback and suggestions. We are gonna address their questions and concerns in the following, and remain available for any additional questions.
>
> **Comparison with [29]**
>
> The method in [29] is one of many recent works [13,18, 20,22, 25, 27] that focus on the emergence of similar representation independently to either measure similarity or solving for a map (implicitly or explicitly) between the two domains.
> The conceptual differences with [29] are in:
> - Solving the alignment problem by optimizing for a map between the eigenbasis of the two domains (functional map). Therefore our framework optimizes for a map between the spaces, while [29] maps both spaces independently to a shared relative space and the mapping is implicit.
> - By employing cosine similarity, [29] assumes that the transformation is a global rotation of the space plus a local rescaling, so that cosine similarity is invariant to the mapping. This means that when there is a more complex transformation [29] will not be able to handle it well, while a functional map can in principle handle complex non linear transformations by expressing the map as a linear sparse matrix in the spectral domain.
> - Our framework comprises at the same time (i) an interpretable similarity measure, (ii) a way to find correspondences, and (iii) solving for an explicit mapping between different spaces in a single framework, differently from [29] which attempts to solve just the latter implicitly.
> We included a discussion in the manuscript.
> In terms of qualitative and quantitative performance we added experiments in Table 2 comparing with [29] in the general Answer, demonstrating better qualitative and quantitative performance.
>
> **Descriptors**
>
> In general in our framework descriptor functions can either be semi supervised (e.g. express on each node of the ) weakly supervised (use the indicator functions of labels ) or fully unsupervised ( unsupervised descriptors of the graph, such as the heat kernel signature[c]).
>
> Neural models, compared to 3D shapes, involve many quantities that can be assumed to be preserved across latent spaces, as indicated in [13, 29, 22, b]. Labels are a clear example, but unsupervised properties related to distributions in latent spaces can also be preserved. For instance, in models that assume a prior distribution in latent spaces, such as Variational Autoencoders and derived models, specific quantities can be conserved and exploited to develop useful descriptors.
>
> We agree that designing better unsupervised descriptors tailored to the data and task at hand is an intriguing direction. This is a key focus area for our future work. We report an ablation study on the choice of descriptors in the General answer.
>
> _[a] Sun, J. et al.. A concise and provably informative multi‐scale signature based on heat diffusion. In Computer graphics forum_
> _[b]Damodaran, B.et al.. Deepjdot: Deep joint distribution optimal transport for unsupervised domain adaptation.ECCV_
>
> **Figure 2**
>
> We adjusted the colorbars in Figure 2 for improved visualization.
>
> **Baselines**
>
> We added a comparison with different baselines [18,25,22] on the CUB dataset on a retrieval task in the general answer. With this experiment, we remark that LFM can be used on top of any off-the shelf method as [18, 22, 25, 27, 20, 13].
>
> We did compare with the method in [22], referred to as “ortho” in Figures 3 and 5 of the manuscript, and in the additional experiments of the general answer. In the main paper we chose to compare mostly with it as it holds the state of art in the stitching and retrieval benchmarks considered.
> With respect to [18,25,22,17] the assumptions are pretty similar, in that they assume a linear mapping and optimize for it.
> We stress that with respect to methods like [13], that compute a similarity kernel   between latent spaces without solving for any mapping, can be easily incorporated in our framework as descriptors of the graph nodes.
>
> Regarding the visualization of stitched autoencoder reconstructions, we have added an experiment using the MNIST, FashionMNIST, and CIFAR-10 datasets. The results, presented in the attached PDF in the general answer section, demonstrate superior reconstruction quality w.r.t. [22].
>
> **Clarity on Figure 3 and 5**
>
> We improved the experimental section stating better the goal and takeaways from each experiment. We will add the update in the camera ready.
>
> **Functional Map visualization**
>
> We added functional maps visualization for experiment in the pdf attached in the general answer in two experiments.
> In Figure, we conduct a synthetic experiment by perturbing the space with simple transformations, such as rotation. We observe that the map remains perfectly diagonal when recovering an isometry of the space. Additionally, we note that it preserves a diagonal-like structure even when noise is added.
> In Figure we show  functional maps visualizations in real case scenarios.
>
> **Comparison with Ortho [22] clarification**
>
> In both Figures 3 and 5 the purpose of comparing between Ortho and LFM + Ortho is to show how our framework can expand the set of correspondences from very few (e.g 3.) to many one and outperform the direct estimation of the Orthogonal mapping by just using the same initial information. We choose to fit the same orthogonal transformation on top of the correspondence found by LFM to make the comparison with [22] the fairest possible, although in principle any off-the-shelf methods could be used to estimate the transformation once the new correspondence is found.
>
> **Variance fine labeling**
>
> The fine labels case is more complex than the coarse label one, as the latter correspond to assign the superclass information to each sample, therefore resulting in having less classes. Therefore estimating the map can be more difficult in some settings. We observed this especially when trying to match self supervised vision models to classification based ones (i.e. DINO vs ViT).

---

> ### Author Response · Authors · 2024-08-11
> **Following up comment on the rebuttal and review**
>
> We thank again the Reviewer for their feedback.
>
> We hope that our rebuttal has adequately addressed their concerns, and we kindly request for feedback on this, particularly in light of the positive responses to the rebuttal from the other Reviewers: 5Rrs, yAi6 and neES.
>
> We remain available to clarify any further questions or concerns during the remainder of the discussion period.

---

> ### Comment · Reviewer_9utW · 2024-08-13
> **Rebuttal Response**
>
> I thank the authors for the rebuttal. Broadly, I am ok with the response and mostly like what I see (Figure 2,4 and Table 2 in the response pdf). I am still concerned with the descriptor problem and, it appears one needs good concrete landmarks to get a good FMap. It is unclear how to generally make such descriptors for representation spaces - fully unsupervised as rightfully pointed out.
>
> I would still critique the authors for a less favorable presentation (even in the rebuttal pdf, there are no captions for each figure and the fonts are too small - making it rather cumbersome to see what issue was being resolved with these additional experiments). Nevertheless, this is an interesting paper that could be compiled much better for long-term impact. Although I am not 100% convinced, I think I can live with a borderline accept score.

---

> > ### Author Response · Authors · 2024-08-14
> > **Response to reviewer's comment**
> >
> > We thank the Reviewer for their feedback and for adjusting their score.
> >
> > Regarding the unsupervised descriptors: we agree that the development of fully unsupervised descriptors is a very relevant direction, we believe it's achievable by encoding information that remains consistent across domains, within a single graph representation.
> >
> > To illustrate this, we tested two unsupervised descriptors on the same descriptor ablation experiment of Figure 4 in the Rebuttal pdf. The new descriptors are computed in the following way:
> >
> > - For the first descriptor, we computed the K-nearest neighbor distances for each node, sorted in increasing order based on the distance function used to build the graph (e.g., cosine distance).
> > - For the second descriptor, we concatenated to the first descriptor to a fraction  of K (specifically 1/10 of K) of the farthest nodes in terms of distance.
> >
> > The goal is to capture both intra-cluster and inter-cluster statistics. We refer to these descriptors as "K-nn (unsupervised)"  and "K-nn + farest (unsupervised)". Their performance, alongside other descriptors, is presented in the table below.
> >
> >
> > |      MRR      | dist_geod (supervised) | dist_cosine (supervised) | labels (weakly supervised)|   Hks (unsupervised)   |  Wks (unsupervised)  | **K-nn (unsupervised)** | **K-nn + farest (unsupervised)** |
> > |:-------------:|:---------:|:-----------:|:------:|:-------:|:------:|:---------------------:|:------------------------------:|
> > | **Value**     |  0.9491   |   0.9491    | 0.9291 |  0.0436 | 0.0111 |        0.6793         |             0.7341             |
> >
> >
> > While these unsupervised methods do not fully match the performance of their supervised counterparts, they achieve strong MRR scores. This demonstrates the feasibility and potential of unsupervised descriptors, which we plan to explore further in future work.
> >
> > We hope this addresses the Reviewer's concerns on this and we remain available for any further clarifications or questions throughout the remainder of the discussion period.

---

### Official Review · Reviewer_neES · 2024-07-10

**Soundness:** 2
**Presentation:** 2
**Contribution:** 2
**Rating:** 6
**Confidence:** 4

**Summary:**

The paper presents a new way to compare neural representations: Latent Functional Maps (LFM). The later one is achieved by 1) building symmetric knn graphs 2) calculating Laplace eigenfunctions 3) calculating optimal mapping between them.
Applications include: (i) compare different spaces in an interpretable way and measure their intrinsic similarity; (ii) find correspondences between them, both in unsupervised and weakly supervised settings, and (iii) to effectively transfer representations between distinct spaces.
Experiments with CNNs and word embeddings are provided.
The language is fine and the manuscript is well organized, but details of the proposed method and experiments are hard to understand.

**Strengths:**

In terms of novelty, to the best of my knowledge, this is a first time when a concept of Latent Functional Mapping is introduced to deep learning. In this paper, it is applied for comparing neural representations.
Developing new scores for comparison of neural embeddings is still an important research topic and could enjoy plantly of applications.
LFM is robust to translations in a directions orthogonal to a separating hyperplane, in opposite to well known CKA.
Experiments include modern ViT architectures.

**Weaknesses:**

1. The notion of Functional Maps is applied "as is", without significant modifications. So, from the theoretical perspective, there is no contribution.
2. Many details of the proposed method and experiment are missing, instead it authors refer to original papers, see below.
It makes the manuscript hard to read. In my opinion, the paper must be self-contained.
Settings of these experiments are not a common knowledge. You can include several small paragraphs introducing them to a reader. Examples:

* line 152: and incorporate regularizers for Laplacian and descriptor operator commutativity, as defined in [31]:
* line 158: in the other latent space. Once we have solved the optimization problem defined in Equation 2, we refine the resulting functional map C using the algorithm proposed by [24].
* line 235: We test the use of the latent functional map in the task of zero-shot stitching, as defined in [29],
* line 185: This functional alignment can be used similarly to the method proposed by [29] to establish a "relative" space where the representational spaces X and Y are aligned.

**Questions:**

1. Is the LFM score symmetric?
2. Is it a metric?
3. What is a descriptor operator? (line 152)
4. What is stitching of latent spaces?
5. Why the method from Moschella et al. is not used for comparison?
6. Why do you use this particular set of functions:
"As a set of corresponding functions, we use the geodesic distance functions"
7. line 145, 146: do you have one-to-one correspondence between X and Y ?

Moschella, L., Maiorca, V., Fumero, M., Norelli, A., Locatello, F., & Rodolà, E. (2022). Relative representations enable zero-shot latent space communication. arXiv preprint arXiv:2209.15430.

**Other:**

I think that the proposed idea is intersting, especially becase it gives not only a similarity score, but tools for scrutiny of embeddings.
But the paper has issues with clarity (see above).

---

> ### Author Rebuttal · Authors · 2024-08-07
>
> We thank the reviewer for their feedback and suggestions. We are gonna address their questions and concerns in the following and remain available for any additional questions.
>
> **Theoretical contribution**
> We would like to respectfully point out that the goal of the paper is to make aware of the representation learning community of the functional map framework, which can have many impactful applications (retrieval, stitching) that were not primarily explored in the geometry processing setting. From the theoretical perspective the dimensionality of the problem is very different.
>
> According to the NeurIPS submission guidelines, combining established methods to address new problems is a valuable contribution. Applying well-studied tools with theoretical guarantees can often lead to significant advancements and insights. In our study, we applied the Functional Map framework with minor adaptations, specifically in graph building within the latent space. Surprisingly, this approach already yielded considerable improvements over previous methods, demonstrating its practical efficacy and potential. We believe these findings are noteworthy and beneficial to the community. Furthermore, our results pave the way for future research, where we plan to explore more tailored modifications of the method.
>
> **Self-containment**
>
> We improved clarity in the paper, in particular:
> line 152: we explained and reported the definition of the operators in Appendix A of the paper in the original submission. We made the link clearer adding references to the main text and details in Appendix. We also incorporated part of it in the main paper to improve the flow of the writing.
> line 158: We reported an in-detail explanation of the zoom-out iterative refinement strategy of [24] in Appendix A. In the main text, we briefly described how the algorithm works and referenced the detailed explanation in the Appendix.
> line 235: We added a self-contained explanation for all experiments, including the definition of zero-shot stitching, which, for clarity, we report below.
> line 185: Adding an in more detailed explanation of the [29], we hope the concept of functional “relative” space is now more clear.
>
> **LFM similarity score**
>
> We regard the LFM similarity score as a pseudometric: concerning symmetry,  as the functional map commutes with the laplacian eigenbases of the two spaces (when optimizing for it, this is enforced by the regularizer in equation 3 in appendix A) . This enforces that the inverse of the functional map will correspond to its transpose. The formula for the similarity is invariant to the transpose operator, making the score symmetric.
> Concerning triangle inequality we believe that the score does not satisfy this property as distinct perturbations applied to space may have similar scores.
> These properties are discussed in the manuscript, and we have included the formal proofs in the Appendix for clarity and completeness.
>
>
> **Descriptor Operator**
>
> As defined in Appendix A, the descriptor operator expresses the set of descriptor functions  \mathbf{F}_{G_X} (likewise \mathbf{F}_{G_Y} ) in eq 2 as an operator in the spectral basis of the graph.  The intuition behind the commutativity constraint in the regularizer eq3 of Appendix A, is for indicator functions of regions of constant values of each pair of descriptors f^{G_X}_i, f^{G_Y}_i to be preserved.
> We expanded the explanation of the definition, and made the reference to the appendix and the text in the main clearer.
>
> **Stitching**
>
> We define a generic stitched model as the composition of an encoder model that embeds data with a decoder specialized in a downstream task (classification, reconstruction). The stitching operation is always performed without training or fine-tuning of the encoder/decoder, in a zero-shot fashion.
> In [29], the notion of zero-shot stitching different neural components was introduced, treating them as frozen black-box modules. Nevertheless we remark that [29] still needed to train a decoder module once in order to process relative representations, before being able to perform stitching, while our method is fully zero-shot.
>
> **Comparison with [29]**
>
> We compare with [29] in the retrieval experiment in Figure 5 of the paper. We added further comparison in the retrieval experiment on the CUB dataset reported in the general answer for this rebuttal. In the stitching experiments, we do not report comparison with [29]  for two reasons: (i) [29] requires training decoder modules once before performing stitching, while we directly estimate the map between the two latent spaces, without requiring any additional training; (ii) We compare with [22] (under the name of ortho), however, whose performance is superior to the one of [29], as reported in the experimental section in [22]. By demonstrating better performance w.r.t. [22] our method is therefore superior to [29].
>
> **Choice of descriptor operators**
>
> In general in our framework descriptor functions can either be semi supervised (e.g. express on each node of the ) weakly supervised (use the indicator functions of labels ) or fully unsupervised ( unsupervised descriptors of the graph, such as the heat kernel signature).
> We explored different strategies for the choice of descriptor functions on top of the KNN graph, and chose the geodesic distance on the graph as it is agnostic of the choice of the metric used to construct the KNN graph  and provides a good choice in terms of performance.
> We included an ablation experiment on the choice of descriptor in the PDF of the general answer. We clarified this in the main manuscript and reported the ablation experiment in the Appendix.
>
> **Correspondence between domains**
>
> In general in our framework we don’t assume to have access to a direct correspondence between the 2 domains, and we can generalize to setting when this information is accessible for some samples (semi supervised) or not available at all (unsupervised).

---

> ### Author Response · Authors · 2024-08-11
> **Following up comment on the rebuttal and review**
>
> We thank again the Reviewer for their feedback.
>
> We hope that our rebuttal has adequately addressed their concerns, and we kindly request for feedback on this, particularly in light of the positive responses to the rebuttal from Reviewers 5Rrs, yAi6.
>
> We remain available for any further questions or concerns during the remainder of the discussion period.

---

> > ### Comment · Reviewer_neES · 2024-08-12
> > **Answer**
> >
> > Thank you for a detailed response. I agree that "combining established methods to address new problems is a valuable contribution", my point was that the paper doesn't contain new theoretical results. I acknowledge the novelty of the paper in a context of deep learning. I think that descriptions of experiments must be more detailed to make reader's task easier.
> > Overall, I'm raising my score.

---

> > > ### Author Response · Authors · 2024-08-13
> > > **Response to Reviewer's comment**
> > >
> > > We thank the Reviewer for their response and for adjusting their score.
> > >
> > > We agree that further exploring the theoretical aspects of latent functional maps would be valuable for future research. A potential starting point could be to better analyze convergence rates of graph Laplacians' spectra to the Laplace-Beltrami Operator's spectra for KNN graphs [5] in our setting, considering that dimensionality and the approximation strategy of the manifold are two key differences in our setting compared to the 2D dimensional manifolds in the original functional map framework.
> > >
> > > We will include a brief discussion of this in the paper’s conclusions and, as discussed, provide comprehensive, self-contained descriptions of the experiments in the main paper.
> > >
> > > We remain available throughout the remainder of the discussion period for any further questions.
> > >
> > > _[5] Calder, J., & Trillos, N. G. (2022). Improved spectral convergence rates for graph Laplacians on ε-graphs and k-NN graphs. Applied and Computational Harmonic Analysis, 60, 123-175._

---

### Official Review · Reviewer_5Rrs · 2024-07-14

**Soundness:** 3
**Presentation:** 1
**Contribution:** 3
**Rating:** 6
**Confidence:** 3

**Summary:**

The paper tackles the problem of modeling relationships between latent spaces learnt by different models. It proposes using functional maps that have been used in 3D vision and graph matching for the purpose. It does so by approximating the latent space structure using a knn graph constructed using anchor points, on which the functional map (LFM) is defined.

LFM is used to stitch together encoders and decoders from separate CNNs trained independently, where it is shown to perform significantly better and with lesser variability than previous work, while simultaneously using less anchors than them.

**Strengths:**

Tackles an important, practically useful problem using a novel approach while showing strong, although limited experimental results.

**Weaknesses:**

**Writing:** The writing quality in the paper could be significantly improved in terms of:

1. Including more details about the experiments in the main manuscript (defining the goal of the experiment formally, detailing backbones, optimizers used in training, specifying which models/layers are stitched). An example of such a lack of detail is Sec 5.2, L235.
2. Describing closely related past works [22], [29] in more detail.
3. Clarity and organization of Sec 3, 4.2.

**Limited Experiments:** The current experimental evaluation seems very limited to models trained on very small datasets, e.g. only CIFAR-100 for the zero-shot stitching despite the relative efficiency of the method. I would like to see if the method performs well on more realistic datasets with more complex latent spaces like ImageNet, or even smaller ones like CUB200.

**Questions:**

I would like to know if the gains in performance due to the method still hold for more practical netowrks that have larger, more complex latent spaces like ImageNet

**Limitations:**

The authors have sufficiently addressed limitations

---

> ### Author Rebuttal · Authors · 2024-08-07
>
> We thank the reviewer for their feedback and suggestions. We are gonna address their questions and concerns in the following and remain available for every additional question.
>
> **Writing quality**
>
> We have significantly improved the manuscript's clarity, making it self-contained and clear. In particular:
> We structured the experimental section by adding details about the settings in the experimental setting and adding a smaller subsection depicting the experiment's goal and takeaway.
> In the experimental section, we added an in-detail description for [29] and [22], making it clear that we are in the most similar setting to have a fair comparison with them.
> We added more details in section 3 and section 4.2, also adding more information in the appendix, complementary to the one already present in Appendix A, to make everything self contained and avoid pointing the reader to past work. Although we thank and agree with the Reviewer that our manuscript is clearer right now, we would like to respectfully point out that Reviewer yAi6 highlighted writing clarity, with confidence 5.
>
> **Limited Experiments**
>
> We included multiple new experiments in the paper with the objective of:
> (i) Validating our stitching performance on multiple datasets (CUB, ImageNet,MNIST, AgNews), including suggested datasets and datasets that lead to more complex latent spaces such as Imagenet;
> (ii) validating our retrieval performance on multiple methods, showcasing the applicability of LFM to state-of-the-art methods;
> (iii) perform experiments on multiple modalities to show how the LFM can be applied successfully to different representations.
> The experiments and their analysis are reported in the attached PDF in the general response.
>
> **ImageNet performance**
>
> As detailed in the general answer, we show stitching experiments on Imagenet, validating the usefulness of the LFM despite the dataset being complex. We would like to respectfully point out that the performance is also dependent on the architecture, for which we employed pre-trained state-of-the-art vision models for all the datasets. The models have been pre-trained with different objectives (classification, self-supervision)  and on multiple modalities (image, text) and can handle complex and large-scale vision and text datasets.

---

> > ### Comment · Reviewer_5Rrs · 2024-08-11
> > **Reply to Rebuttal**
> >
> > I have gone through the rebuttal and other reviews, and it takes care of my concerns on experiments, so I increase my score.

---

> > > ### Author Response · Authors · 2024-08-11
> > > **Response to Reviewer's comment**
> > >
> > > We thank the reviewer for their response and for adjusting their score. We remain available for any further questions during the remainder of the discussion period.

---

### Author Rebuttal · Authors · 2024-08-07

We thank all the reviewers for their feedback and suggestions. In the following we addressed some general comments raised by the reviewers and attached a pdf with the experiments performed during the rebuttal period. We remain available for any further question or clarification during the discussion period.

## Additional Experiments

We significantly extended our experimental analysis, reporting the results in the attached pdf and explaining the experiments performed in the following:


**Different datasets**

We benchmarked our method on multiple datasets (ImageNet [a], Mnist [b], AGNews [c],  Caltech-UCSD Birds-200 [e]) on stitching task in Figure 1. This includes large scale datasets with complex structure such as ImageNet, and multiple modalities (text and images). The stitching experiments confirmed how LFMAP is consistently superior in all these settings to the state of the art baseline of [22], especially when the number of known correspondences is low or absent (labels descriptor case).

**Multiple baselines**

In Table 2 we added a comparison with multiple baselines  [18,25,22,29,f] on the Caltech-UCSD Birds-200-2011 (CUB) dataset on retrieval tasks with the same foundation models and settings used in the main paper. We demonstrate that LFM is consistently superior in performance and can be used on top of any method which computes an explicit mapping between spaces.

**Qualitative evaluation**

In Figure 2 we added qualitative experiments on the MNIST, FashionMNIST and CIFAR 10 datasets,  concerning visualizing reconstructions of stitched autoencoders. For these experiments we trained convolutional autoencoders with two different seeds, and stitched their encoder decoder module. For our method and [22] we used 10,10, and 50 correspondences for the  MNIST, FashionMNIST [g] and CIFAR 10 datasets, respectively. We observe consistently superior reconstruction quality w.r.t. [22] and the absolute baseline.

**Choice of Descriptors**

In Figure 4, we performed an ablation study on the choice of descriptors, comparing supervised descriptors (geodesic distance and cosine distance using 10 correspondences), weakly supervised descriptors (label descriptor), and fully unsupervised descriptors (heat kernel signature [i] and wave kernel signature [h]).
To conduct this study, we performed a retrieval task on the test embeddings of two convolutional autoencoders trained on MNIST, which differed by their parameter initialization. We visualized the structure of the functional map and reported the performance in terms of mean reciprocal rank (MRR), observing the following: (i) Geodesic and cosine descriptors performed best (ii) The geodesic distance (shortest path) is a good choice as it is agnostic of the metric chosen to build the initial graph, yet provides the same result as using the metric itself; (iii) The structure of the functional map reflects the performance of retrieval.

**Functional map structure**

In Figure 3 and Table 1, we present a visualization of the structure of the functional map in a synthetic setting. The experiment was conducted as follows: given an input set \(X\) consisting of test embeddings extracted from MNIST, we aimed to observe the degradation of the functional map structure as the space is perturbed. The perturbation involved an orthogonal transformation combined with additive Gaussian noise at increasing levels. In the first row, the functional maps were computed from k-nearest neighbor (knn) graphs using the cosine distance metric, while in the second row, knn graphs were constructed using the Euclidean distance metric. Below each functional map, the LFM similarity score and MRR retrieval scores are displayed. We observed that (i) when noise is absent (first column), the two spaces are isometric, and the functional map is diagonal, (ii) constructing the graph with the cosine distance metric is more robust to increasing noise, and (iii) the LMF similarity score correlates with the MRR retrieval metric, indicating that more structured functional maps reflect better alignment between spaces.

## Clarity

We have significantly improved the manuscript's clarity, making it self-contained and clear. In particular:
We structured the experimental section by adding details about the settings in the experimental setting and adding a smaller subsection depicting the experiment's goal and takeaway.  In the experimental section, we added an in-detail description for [29] and [22], making it clear that we are in the most similar setting to have a fair comparison with them. We added more details in section 3 and section 4.2, also adding more information in the appendix, complementary to the one already present in Appendix A, to make everything self contained and avoid pointing the reader to past work. Although we thank and agree with the Reviewer that our manuscript is clearer right now, we would like to respectfully point out that Reviewer yAi6 highlighted writing clarity, with confidence 5.

_[a] Deng, J. et al., 2009. Imagenet: A large-scale hierarchical image database.IEEE_

_[b] Deng, L.,. The mnist database of handwritten digit images for machine learning research. IEEE_

_[c] Zhang,et al "Character-level convolutional networks for text classification." Advances in neural information processing systems 28 (2015)_

_[e] Wah,et al The Caltech-UCSD Birds-200-2011 Dataset_

_[f] Conneau, A., et al,. Word translation without parallel data, ICLR_

_[g] Xiao, H.; Rasul, K. & Vollgraf, R. (2017), Fashion-MNIST: a Novel Image Dataset for Benchmarking Machine Learning_

_[h] Aubry, M., Schlickewei, U., & Cremers, D. (2011, November). The wave kernel signature: A quantum mechanical approach to shape analysis. In 2011 IEEE international conference on computer vision workshops (ICCV workshops) (pp. 1626-1633). IEEE_

_[i] Sun, J. et al.. A concise and provably informative multi‐scale signature based on heat diffusion. In Computer graphics forum_

---

### Decision · Program_Chairs · 2024-09-25

**Decision:**

Accept (poster)

**Comment:**

### Summary
This paper proposes a new approach for comparing neural representations by approximating the associated manifold with a kNN graph and utilizing functional maps, a technique that leverages the spectral properties of the manifold to compute relationships between distinct representations, such as similarity.



$$$$

### Strengths
- **Significance**: The paper tackles an interesting and practically relevant problem. The proposed approach presents a novel combination of existing techniques in a simple and elegant manner, which could inspire future research.
- **Experiments**: The experimental results effectively demonstrate the efficiency of the proposed approach.

$$$$

### Concerns
- **Clarity**: The writing could be improved to make the paper more self-contained and accessible to non-experts. Some aspects of the method and the experimental setup should be explained more explicitly. Additionally, a more thorough discussion of related work would be beneficial.

- **Limitations**: The paper does not address certain technical limitations, particularly those related to partial manifolds and the design of descriptors in general settings.

$$$$

### Overall
Considering the contributions of the paper -- particularly its simple yet effective modeling approach and potential impact -- along with the efforts made to address the reviewers' questions and the additional experiments provided in the rebuttal, I believe this paper is a reasonable contribution to the conference and recommend acceptance. While the clarity was criticized, the requested improvements can be implemented, and a discussion of the limitations can be included.